# Biodiversity and Bioprospecting of Fungal Endophytes from the Antarctic Plant *Colobanthus quitensis*

**DOI:** 10.3390/jof8090979

**Published:** 2022-09-19

**Authors:** Laura Bertini, Michele Perazzolli, Silvia Proietti, Gloria Capaldi, Daniel V. Savatin, Valentina Bigini, Claudia Maria Oliveira Longa, Marina Basaglia, Lorenzo Favaro, Sergio Casella, Benedetta Fongaro, Patrizia Polverino de Laureto, Carla Caruso

**Affiliations:** 1Department of Ecological and Biological Sciences, University of Tuscia, Largo dell’Università, 01100 Viterbo, Italy; 2Centre Agriculture Food Environment (C3A), University of Trento, Via E. Mach 1, 38098 San Michele all’Adige, Italy; 3Research and Innovation Centre, Fondazione Edmund Mach, Via E. Mach 1, 38098 San Michele all’Adige, Italy; 4Department of Agriculture and Forest Sciences, University of Tuscia, Via S. Camillo de Lellis, 01100 Viterbo, Italy; 5Department of Agronomy Food Natural Resources Animals and Environment, University of Padova, Viale dell’Università 16, Legnaro, 35020 Padova, Italy; 6Department of Pharmaceutical and Pharmacological Sciences, University of Padova, Via F. Marzolo 5, 35131 Padova, Italy

**Keywords:** Antarctic fungi, DNA barcoding, culturomics, extracellular enzymes, bioactive compounds, plant–fungus interaction

## Abstract

Microorganisms from extreme environments are considered as a new and valuable reservoir of bioactive molecules of biotechnological interest and are also utilized as tools for enhancing tolerance to (a)biotic stresses in crops. In this study, the fungal endophytic community associated with the leaves of the Antarctic angiosperm *Colobanthus quitensis* was investigated as a new source of bioactive molecules. We isolated 132 fungal strains and taxonomically annotated 26 representative isolates, which mainly belonged to the Basidiomycota division. Selected isolates of *Trametes* sp., *Lenzites* sp., *Sistotrema* sp., and *Peniophora* sp. displayed broad extracellular enzymatic profiles; fungal extracts from some of them showed dose-dependent antitumor activity and inhibited the formation of amyloid fibrils of α-synuclein and its pathological mutant E46K. Selected fungal isolates were also able to promote secondary root development and fresh weight increase in Arabidopsis and tomato and antagonize the growth of pathogenic fungi harmful to crops. This study emphasizes the ecological and biotechnological relevance of fungi from the Antarctic ecosystem and provides clues to the bioprospecting of Antarctic Basidiomycetes fungi for industrial, agricultural, and medical applications.

## 1. Introduction

Antarctica is one of the last pristine environments on Earth and represents a valuable area for bioprospecting [1]. Its microbial diversity is largely unexplored and potentially encloses microorganisms endowed with activities of high relevance for white, green, and red biotechnologies [2]. To enable growth under the extreme environmental conditions of Antarctica, microbes evolved several adaptations strategies which are often accompanied by modifications to both gene regulation and metabolic pathways, increasing the possibility of finding unique functional metabolites of biotechnological importance. These adaptations include cellular storage of cryoprotectants such as sugars and polyols to maintain turgor pressure [3], high levels of unsaturated membrane phospholipids to stabilize membranes [4], and fungal melanin for protection against freezing and UV radiation [5], and the production of antifreeze proteins as well as cold-active enzymes [6]. Thus, microbial extremophiles turn out to be a chemical reservoir of extremolytes and extremozymes that have the potential to be outstanding resources for the development of a bio-based economy [2,7]. Extremozymes are enzymes that have evolved molecular mechanisms of adaptation to extreme physicochemical conditions and might have relevant applications in industrial biotransformation processes [2]. They naturally exhibit features that are usually gained by means of laborious synthetic biology approaches or can be used as a scaffold to design tailor-made biocatalysts for application in eco-friendly industrial processes [8]. Most extremozymes characterized from Antarctic filamentous fungi and yeasts are classified as hydrolases, but some oxidoreductases, such as laccase and superoxide dismutase, have also been reported [9] and references therein. A thorough study by Loperena et al. [10] reported the screening of extracellular enzymatic activities from 120 bacteria, 31 yeasts, and 10 filamentous fungi isolated from different environmental samples of Maritime Antarctica, highlighting that the most frequent and highest activities were from hydrolases. In particular, amylase, caseinase, lipase, and gelatinase activities were mainly detected in bacterial isolates, whereas yeasts and filamentous fungi more frequently showed lipase and cellulase activities, and little or no detectable pectinase, xylanase, or amylase activities [10].

Besides extremozymes, microorganisms living under stressful environmental conditions are also a valuable source of extremolytes and secondary metabolites, which can constitute up to 25% of dry cell weight [2,7]. The possible commercial applications of extremolytes include anticancer drugs, antioxidants, cell-cycle-blocking agents, and sunscreens, among others [11]. Tian and colleagues [12] reviewed 219 novel natural products from the Arctic and the Antarctic microorganisms, lichen, moss, and marine faunas endowed with antimicrobial, antiviral, and cytotoxic activities towards numerous cell lines as well as free radical scavenging properties ([12] and references therein). In particular, new asterric acid derivatives, peptaibols, epipolythiodioxopiperazines, polyketides, terpenes and sesquiterpenes endowed with multiple biological activities were isolated from Antarctic soil and deep-sea derived fungi [12]. Moreover, extracts from filamentous fungi coming from different Antarctic environments were found to be active against *Xanthomonas* phytopathogens [13]. The biological activity of secondary metabolites from endophytic fungi associated with the only two endemic vascular plants of Maritime Antarctica, i.e., *Deschampsia antarctica* Desv. and *Colobanthus quitensis* (Kunth) Bartl, is less characterized and, to date, only a few fungal extracts from endophytes of these plants were found to possess Leishmanicidal and antitumoral activities [14]. In addition to the importance of extremophiles in bioprospecting for bioactive molecules, they also exert a pivotal role in modulating plant and animal performance in stressful conditions. Such organisms are inhabited by microorganisms forming metaorganisms, whose functionality is strictly dependent by the dynamic crosstalk between the community components [15]. Interestingly, this kind of functional symbiosis has also been reported for the two Antarctic angiosperms [16,17]. *C. quitensis* leaves metatranscriptome disclosed transcripts belonging to fungi, bacteria, viruses, and mycoviruses revealing the holobiontic nature of the plant [18,19,20]. The microbial communities associated with both Antarctic vascular plants have been widely reported to enhance their tolerance to stress, improving the uptake of nutrients and water, and the resistance to UV-B radiation, drought, saline stress, and simulated global warming [20,21,22,23,24,25,26]. Interestingly, Antarctic plant symbionts were also found to improve the ecophysiological performance of agricultural species under abiotic stress conditions [27,28]. Thus, great interest is growing around the possibility to exploit extremophilic microbes as a biotechnological tool for a sustainable agriculture, even though only a few reports are available to date regarding the use of extremophiles as potential biocontrol agents for crop disease management and as potential enhancers of crop productivity [29]. With the advent of the genomic era, genome and metagenome mining technologies have been progressively applied to discover genes that are likely to produce novel bioactive compounds [30]. However, culture-dependent isolation techniques, followed by bioactivity-guided fractionation and identification of purified compounds, are still necessary for applied studies [9] and indispensable for the molecular and functional characterization of microorganisms.

In this study, we identified fungal endophytes from the leaves of the Antarctic angiosperm *C. quitensis*. Moreover, selected fungal isolates have been used as a source to identify metabolites and enzymes for bioprospecting purposes. In addition, plant–fungus interaction experiments were carried out on model plants with the aim of evaluating their possible use as biocontrol agents and/or plant growth promoters in eco-friendly solutions for sustainable agriculture.

## 2. Materials and Methods

### 2.1. Samples Collection

Sampling was carried out in the proximity of the Henryk Arctowski Antarctic Research station, King George Island, Maritime Antarctica (62°14′ S, 58°48′ W). *Colobanthus quitensis* plantlets were collected from three sites (S1, 62°9′44.58″ S, 58°27′58.68″ W; S2, 62°9′49.62″ S, 58°28′7.02″ W; S3, 62°9′52.90″ S, 58°28′21.31″ W) differing in the distance from the coastline, soil composition, altitude, temperature, and wind exposure. Besides collecting plant samples grown in open areas (OA samples), plantlets were also collected inside small greenhouses open on the top (Open Top Chambers, OTCs), placed only in S2 and S3, which determine an increase of about 4 °C during midday, mimicking the effect of global warming (OTC samples). Following sampling, plants were placed in 2 mL cryo-vials filled with 0.2 mL of sterile water and stored at 4 °C until isolation of fungal endophytes that was performed at the arrival of the samples in Italy.

### 2.2. Isolation of Culturable Fungal Endophytes

*C. quitensis* plants were surface-disinfected as described in [31] with 70% ethanol for 1 min, 2% sodium hypochlorite for 1.5 min, and 70% ethanol for 1 min. Plants were washed three times with sterilized distilled water (2 min each) and samples (0.5 g) were ground in stainless jars using a mixer-mill disruptor (MM 400, Retsch, Haan, Germany) at 25 Hz for 2 min in the presence of 1 mL 0.85% NaCl. Culturable fungi were isolated as previously described [32] by plating serial dilutions of each suspension (100 µL aliquots) on Potato Dextrose Agar (PDA) (Oxoid Ltd., Basingstoke, Hampshire, UK) supplemented with 0.25% lactic acid to inhibit bacterial growth [33,34]. As control of plant surface sterilization, aliquots (100 µL) of the last washing solution were plated to confirm the absence of microbial growth. Plates were incubated at 15 ± 1 °C and 25 ± 1 °C and fungal colony-forming units per gram of plant fresh weight (CFU g^−1^) were assessed daily for up to 60 days [35]. Three biological replicates were analyzed for each of five conditions (S1.OA, S2.OA, S2.OTC, S3.OA, S3.OTC) and two technical replicates were performed for each sample. The absence of bacterial growth was confirmed on all plates according to colony morphology and microscopic analyses.

### 2.3. Identification and Selection of Fungal Endophytes

Colony morphology and microscopic characteristics were examined starting from the eighth day after inoculation on PDA plates incubated at 25 °C. Fungal isolates were initially divided into different ‘morphotypes’ based on morphological features such as colony texture, degree of sporulation, obverse and reverse colony colors, production of soluble pigments and exudates. Sporulating fungi were identified based on the morphological characteristics of the conidiophores and conidia after fungal material was mounted in lactophenol, cotton blue or acid fuchsin for microscopical observation under the Eclipse 80i microscope (Nikon Inc., Melville, NY, USA). The isolates were identified through their cultural and morphological profiles compared with available literature [36,37]. To reduce redundancy, one representative isolate for each morphotype was selected and taxonomically annotated by amplification of the fungal internal transcribed spacer (ITS) region with the DreamTaq DNA Polymerase (Thermo Fisher Scientific, Waltham, MA, USA) using specific primer pairs (ITS1f: 5′-TCCGTTGGTGAACCAGCGG-3′ and ITS4r: 5′-TCCTCCGCTTATTGATATGC-3′). PCR products were purified by the NucleoSpin PCR Clean-up purification kit (Macherey-Nagel, Dueren, Germany) and sequenced with an ABIPRISM 3730xl DNA analyzer (Applied Biosystems, Thermo Fisher Scientific). ITS sequences were deposited in the NCBI database (http://www.ncbi.nlm.nih.gov; accessed on 1 June 2022) (Appendix A). ITS sequences were aligned (BLASTn) against the nucleotide ITS database of fungi type and reference material of the NCBI (http://www.ncbi.nlm.nih.gov; accessed on 1 March 2022) and genus rank annotation was adopted when equal BLAST top score similarity values (ranging from 96 to 100% of sequence identity) were obtained for species belonging to the same genus, according to [38]. The phylogenetic tree was constructed on ITS sequences with the two best BLASTn hits using the neighbor-joining method and a bootstrap analysis with 1000 replicates with MEGA version 7.0 [39].

### 2.4. Fungal Growth and Samples Preparation

Fungal isolates were grown at 25 °C on PDA plates or in 100 mL flasks containing 30 mL Potato Dextrose Broth (PDB, Oxoid Ltd., Basingstoke, Hampshire, UK) inoculated with 5 mm agar plugs cut from the margin of fungal colonies developed on PDA plates. The flasks were incubated at 25 °C under shaking (150 rpm) in the dark. After 3–4 weeks, the cultures were aseptically centrifuged at 5000 rpm for 15 min to obtain cell-free supernatants; pellets were washed twice with sterile NaCl 0.9% and both supernatants and pellets were lyophilized. The lyophilized supernatants were resuspended in one tenth of sterile deionized water and used for antibacterial activity evaluations and enzymes in vitro assays. Pellets were used for metabolites extraction as described below.

### 2.5. Antibacterial Activity Evaluation

The antimicrobial properties of concentrated supernatants from the newly isolated endophytic fungi were assessed by the agar well diffusion method against a range of pathogens and model microorganisms. Bacterial isolates used for screening, included Grampositive and Gram-negative microorganisms (*Listeria innocua* 652, *Listeria monocytogenes* ATCC 19117, *Bacillus cereus* ATCC 11778, *Staphylococcus aureus*, *Pseudomonas aeruginosa* PAO 1 GFP, *Escherichia coli* APEC 13422_1_CER78, *Escherichia coli* APEC 18042/2, *Salmonella enteritidis*, *Salmonella gallinarum*, *Enterobacter* sp. CCH15). Bacterial isolates were grown in Nutrient Broth at 37 °C under shaking (200 rpm). The 24 h cultures were diluted to 10^6^ CFU mL^−1^ with sterile NaCl 0.9%. One-hundred microliters of the bacterial suspensions were spread on nutrient agar plates used for the well diffusion assays; plates were maintained at room temperature while digging 5 mm diameter wells in the agar using a sterile glass pipette. Wells were then filled with 50 μL of concentrated supernatants buffered at pH 7 with NaOH. The plates were maintained for about 1 h at room temperature to allow diffusion and then incubated at 37 °C for 24 h before measuring inhibition zone diameters. Triplicate plates were prepared for each fungal sample and bacterial isolate.

### 2.6. Extracellular Enzyme Activities

The amylolytic, lipolytic, cellulolytic, pectinolytic, and proteolytic activities of fungi were evaluated on solid media, added with specific substrates, placing 5 mm fungal plugs on the plates, as described in [40]. After incubation for 7–15 days at 25 °C, the appearance of haloes, indicating enzyme activity around the fungal colonies, was observed. Proteolytic activity was assessed using both skimmed milk and gelatin as substrates. Isolates exhibiting the highest extracellular amylolytic and cellulolytic production based on their hydrolysis haloes were selected and enzymatic activities on starch and carboxymethylcellulose (CMC) were measured in the concentrated supernatants at different pH, i.e., 3, 4, 5, 6, 7, as described in [40,41]. Enzymatic activities were expressed as nanokatals mL^−1^ (nKat mL^−1^), defined as the enzyme activity needed to produce 1 nmol of glucose per second per mL of culture.

### 2.7. Fungal Metabolite Extraction

Metabolite extraction was carried out from fungal isolates of *Trametes* sp. S2.OA.C_F6, *Lenzites* sp. S3.OA.B_F6 and *Sistotrema* sp. S1.OA.C_F2 according to [42,43]. The lyophilized fungal culture (200 mg) was treated with 60% cold methanol containing 1% formic acid (5 mL). After 30 s, the sample was frozen in liquid nitrogen for 5 min and then thawed on ice. Methanol extracts were centrifuged twice at 13,000 rpm for 15 min at 4 °C and the pellet was re-extracted. The water extracts were obtained by dissolving 200 mg of lyophilized fungal culture in 10 mL of hot water (80 °C) for 2 h and the solution was filtered on Whatman filter paper. Both extracts were lyophilized and stored at −80 °C. The samples were resuspended in the appropriate buffer depending on the experiments just before the analysis.

### 2.8. Determination of Total Phenolic and Total Flavonoid Content

Total phenolic content (TPC) of extracts was obtained according to Folin–Ciocalteu’s method [44,45]. All analyses were carried out with a Perkin Elmer Lambda-25 spectrophotometer (Shelton, CT, United State). The absorbance was measured at 765 nm. TPC was expressed as mg of Gallic acid equivalents (GAE) g^−1^ dry weight [46,47]. The total flavonoid content (TFC) was obtained by using the aluminum chloride colorimetric method [48,49]. For each sample, 30 μL of fungal extract (40 mg mL^−1^, dry weight) was mixed with 120 μL of absolute MeOH, 6 μL of 0.75 M AlCl3, 6 μL of 1 M CH3COONa, and 170 μL H_2_O. After incubation at room temperature for 40 min, the mixture absorbance was measured at 415 nm. TFC was expressed as mg of quercetin equivalents (QE) g^−1^ dry weight.

### 2.9. Cell Viability Assay

Primary cultures of human lung epithelial HBEC3-KT (ATCC^®^ No. CRL-4051™) and neuroblastoma SH-SY5Y (ATCC^®^ No. CRL-2266™) cells were cultured at 37 °C under 5% CO_2_ for 2 days in 75 cm^2^ flasks (NUNC, Roskilde, Denmark). The flasks were pre-treated with gelatin and a Dulbecco’s Modified Eagle Medium/Nutrient Mixture F-12 (DMEM-F12) supplemented with 10% fetal bovine serum, 1% penicillin-streptomycin, 1% glutamine, and 0.8% human epidermal growth factor. Confluent cells were detached using a trypsin-EDTA solution and then sedimented by centrifugation at 100× *g* for 5 min. After removing the supernatant, the cell pellets were resuspended in fresh medium, counted in a Bürker Counting Chambers, and seeded at 5000 cells per well in 96-well plates (NUNC, Roskilde, Denmark). The Viability assay with PrestoBlue reagent was performed according to the manufacturer’s protocol (Invitrogen, Thermo Fisher Scientific, Waltham, MA, USA). After exposure of the cells to 100 μL of fungal extracts at increasing concentrations (0.5, 1, and 5 mg mL^−1^) for 24 and 48 h, the viability was detected by recording the absorbance at 570 nm by using a VICTOR^®^ Nivo™ system. The cellular viability was expressed as the percentage of viability as related to untreated cells.

### 2.10. Expression and Purification of Recombinant Human α-Synuclein

Human α-Synuclein (Syn) and its mutant E46K were expressed in *E. coli* BL21 (DE3) cell line transfected with the pT7–7/α-syn plasmid [50]. Overexpression of the protein was achieved by growing the cells in LB medium at 37 °C to an absorbance at 600 nm of 0.6, followed by induction with 0.5 mM isopropyl-thiogalactopyranoside. The purification of the recombinant protein was carried out following a previously described procedure [50].

### 2.11. Aggregation of α-Synuclein In Vitro

Recombinant lyophilized protein was dissolved in 20 mM sodium phosphate buffer, pH 7.4, and then filtered with a 0.22 µm PVDF membrane (Millipore, Bedford, MA, USA). Protein concentration was estimated by UV spectroscopy using the molar absorption at 280 nm for Syn and E46K samples (ε_M_ = 5960 M^−1^ cm^−1^). The final concentration was 70 µM (1 mg mL^−1^). The fibril formation assay was carried out in an Eppendorf Thermomixer Compact (Eppendorf, Hamburg, Germany) incubating the samples for up to 7 days (168 h) at 37 °C under shaking at 750 rpm in the absence and in the presence of *Trametes* sp. S2.OA.C_F6 methanol extract (1:10, Syn (or E46K)/extract, *w*/*w*). Aliquots were collected at 0, 72, and 168 h and analyzed by the Thioflavin T (ThT) binding assay [51] and far UV circular dichroism (CD) measurements. A Cary Eclipse fluorescence spectrophotometer (Agilent Technologies, Santa Clara, CA, USA) was used to measure the dye emission in the 455–600 nm range, after excitation at 440 nm, of a freshly prepared 25 µM ThT solution in 25 mM sodium phosphate buffer, pH 6.0, filtered with a 0.22 µm PES membrane, at a final Syn (or E46K) concentration of 60 µg mL^−1^. Far-UV CD spectra were recorded on a Jasco (Tokyo, Japan) J-710 spectropolarimeter, using a 1.0 mm path length quartz cell and a protein concentration of 7 μM. The mean residue ellipticity [θ] (degree cm^2^ dmol^−1^) was calculated from the formula [θ] = (θ_obs_/10) (MRW/lc), where θ_obs_ is the observed ellipticity in degrees; MRW is the mean residue molecular weight of the protein; *l* is the optical path length in cm, and *c* is the protein concentration in g ml^−1^. The spectra were recorded in PBS. Transmission Electron Microscopy (TEM) was performed by negative staining method, in which 5 µL of the sample were diluted to 0.25 mg mL^−1^ of protein; the sample was dried on the plastic grid and subsequently stained with 10 µL of 1% (*w*/*v*) uranyl acetate solution. A Tecnai G2 12 Twin TEM microscope (FEI Company, Hillsboro, OR, USA) was used for sample imaging.

### 2.12. Plant Growth Conditions and Plant-Fungi Co-Cultivation Experiments

Tomato seeds (*Solanum lycopersicum* L. cv. AKRAI F1) were surface sterilized with 95% (*v*/*v*) ethanol for 5 min followed by incubation in 1% (*v*/*v*) sodium hypochlorite for 7 min. *Arabidopsis thaliana* ecotype Columbia 0 (Col-0) were surface sterilized for 10 min, under gentle shaking with 0.08% (*v*/*v*) sodium hypochlorite containing 0.01% SDS. Following sterilization, seeds were washed 3 times with sterile water and then placed on Petri dishes (90 mm diameter) containing half-strength MS nutrient medium (2.245 g L^−1^ MS powder) [52]. The medium was supplemented with sucrose 0.5% (*w*/*v*) (Sigma-Aldrich, Burlington, MA, USA) and solidified with agar 0.8% (*w*/*v*) (Duchefa Biochemie, Haarlem, The Netherlands) at pH 5.8. Tomato seed plates were incubated vertically in a growth chamber at 23 °C with a 16 h light/8 h dark photoperiod, whereas Arabidopsis seeds were first vernalized overnight at 4 °C, under dark conditions, and then moved to a growth chamber at 24 °C with a 14 h light/10 h dark photoperiod. When all seeds were germinated, seedlings of the same dimension were transferred on Petri dishes of 150 mm diameter. Two fungal plugs of 5 mm diameter, cut from the growing edge of *Trametes* sp. S2.OA.C_F6 and *Lenzites* sp. S3.OA.B_F6 cultures on PDA plates (Duchefa Biochemie, Haarlem, The Netherlands), were placed at a distance of 5 cm from the roots, and co-cultivation experiments were protracted for 3 days for tomato and 7 days for Arabidopsis. In half of the plates, a furrow was made using a sterile scalpel to physically separate the seedlings from contact with the fungus. At the end of the exposure period, fresh weight and root length were measured for each seedling. In the control experiments, plantlets were grown in the absence of the fungus. Plate digital photographs were taken at a resolution of 600 dpi (dots per inch) with an Epson Expression V800 Photo scanning system. The primary root length of all plantlets was measured from the starting point of the root differentiation zone and quantified by using the freely available ImageJ software (http://rsbweb.nih.gov/ij/, accessed on 10 February 2022).

### 2.13. Co-Cultures of Antarctic Fungi and Fungal Phytopathogens

The interaction between *Trametes* sp. S2.OA.C_F6, *Lenzites* sp. S3.OA.B_F6, *Peniophora* sp. S3.OTC.C_F1 and *Sistotrema* sp. S1.OA.C_F2 with the phytopathogens *Fusarium graminearum* and *Botrytis cinerea* was tested in vitro. All fungal isolates were grown on PDA medium in 90 mm Petri dishes, and their growth rate was recorded measuring the mycelium radial growth. For co-cultivation experiments, a 5 mm diameter fungal plug, cut from the growing edge of each Antarctic fungal isolate, was placed on PDA medium (90 mm Petri dishes) on the opposite side from the pathogen. The plates were sealed with parafilm and incubated at 24 °C for 7 days in the light. The same co-cultivation experiment was carried out on PDA medium supplemented with bromophenol blue 0.005% (*v*/*v*), as a pH indicator (yellow, pH < 3.5; blue, pH > 4.6). The formation of a yellow halo within the bromophenol blue medium indicates the acidification of the medium induced by the mycelium.

### 2.14. Statistical Analysis

All data were reported as mean values ± standard deviation (SD) of biological triplicates. Statistical analysis was carried out using one-way ANOVA with a *p*-value < 0.05 regarded as statistically significant. In the analysis the *p*-values are indicated as * <0.05, ** <0.01, *** <0.001. The differences between control and experimental samples were determined by t-test (Microsoft Excel program for Windows, v.2005 and Origin 9) or Tukey post hoc test (GraphPad Prism 7.0 Software Inc., San Diego, CA, USA), as indicated in the figure legends.

## 3. Results and Discussion

### 3.1. Isolation and Identification of Endophytic Fungi from the Antarctic Plant Colobanthus quitensis

A total of 132 endophytic fungi were isolated only from the plates incubated at 25 ± 1 °C with no CFU differences among the collection sites (Figure 1A), and no fungal CFUs were found during incubation at 15 ± 1 °C. The absence of fungal growth at 15 ± 1 °C suggests that growth conditions of Antarctic isolates require further optimizations to allow their growth on artificial media in vitro. For example, additional growth media, possibly containing *C. quitensis* plant extracts or supplementary nutritional factors, are required to improve the growth of *Colobanthus quitensis* endophytes in vitro and to better assess their performance at low temperatures. On the other hand, it should be also considered that Antarctic endophytic fungi may be protected by the plants from the direct exposure to the harsh environmental conditions, growing at temperatures slightly higher than the outside. Thus, the optimal growth temperature could be different than expected. Taxonomic annotation of fungal isolates by morphological analysis revealed the dominance of *Trametes* sp., *Sistotrema* sp., *Lenzites* sp., and *Penicillium* sp. (Figure 1B). To reduce redundancy for the functional characterization, 26 representative fungal isolates were selected according to the colony characteristics. The sequence of their ITS confirmed the dominance of *Trametes* sp. (seven isolates), *Sistotrema* sp. (seven isolates), *Lenzites* sp. (two isolates), *Penicillium* sp. (two isolates), *Peniophora* sp. (two isolates) and highlighted the presence of one isolate of *Cladosporium* sp., *Fusarium* sp., *Leucosporidium* sp., *Mollisia* sp., *Phlebia* sp., and *Ypsilina* sp. (Appendix A and Appendix A). Although the possibility of having isolated siblings from the replicas cannot be totally ruled out, it seems unlikely due their differential enzymatic activities found among the selected endophytic fungal isolates (see Section 3.2 below).

Ascomycota is the most abundant phylum of culturable fungi isolated from *C. quitensis* samples in Antarctica [14,35,53]. In particular, *Penicillium* sp. has been identified as one of the dominant fungal genera of *C. quitensis* rhizosphere, roots, and leaves, indicating that cosmopolitan cold-adapted taxa can establish ecological relationships with their plant host [54]. Moreover, taxa phylogenetically near to fungi that can cause diseases in plants and animals, such as *Fusarium*, *Cladosporium*, *Botrytis*, and *Alternaria*, have been previously isolated from Antarctic plants [35,55], corroborating their possible functional roles on *C. quitensis*. Filamentous Basidiomycetes have rarely been isolated from plants inhabiting the Antarctic Peninsula and even their typical role in wood decay has been acquired by Ascomycetes in Antarctica [56]. More commonly, Basidiomycetes have been isolated from the Antarctic soil, such as *Sistotrema brinkmannii* that was isolated from the Antarctic Dry Valleys [56], whence other Basidiomycetes are being recorded by molecular approaches [57]. This indicates that further studies are required to better characterize the distribution and frequency of these fungal taxa in Antarctica. Basidiomycota members were often detected in root-associated endophytic fungal communities from Arctic regions [58]. Among the Basidiomycetes isolated in this study, *Phlebia* sp., *Trametes* sp. and *Peniophora* sp. are known to cause a white-rot type of wood decay, and they have been reported as endophytes from a variety of hosts across wide geographic locations [59], suggesting that further functional and genomic characterizations (e.g., sequencing of more than one genetic marker) are required to improve the taxonomic annotation at the level of species and to verify the existence of endemic taxa adapted for the association with herbaceous Antarctic plants. Pegler et al. [60] reported the presence of *Trametes versicolor* on introduced wood brought to South Georgia, indicating that the possible introduction of non-endemic taxa should be considered as well. However, other decomposer Basidiomycetes can presumably exist naturally in bryophyte and grass ecosystems [61].

### 3.2. Antimicrobial and Enzymatic Activity of Selected Newly Isolated Endophytic Fungi

Among the representative fungal isolates, eight were selected according to the taxonomic annotation and collection site to be screened for the production of industrially relevant extracellular enzymes. Specifically, we selected four isolates of *Trametes* sp., two of *Lenzites* sp., one of *Sistotrema* sp., and one of *Peniophora* sp. (Table 1). The majority of the tested fungal isolates exhibited hydrolytic activities on starch, lipid, cellulose, pectin, and proteins (Table 1), and they showed no antibacterial activity against strains of *Listeria innocua*, *Listeria monocytogenes*, *Bacillus cereus*, *Staphylococcus aureus*, *Pseudomonas aeruginosa*, *Escherichia coli*, *Salmonella enteritidis*, *Salmonella gallinarum*, and *Enterobacter* sp.

*Peniophora* sp. S3.OTC.C_F1 produced hydrolysis halos only on cellulose. On the other hand, the isolates of *Trametes* sp. displayed broad extracellular enzymatic profiles. Lipolytic and pectinolytic enzymes were common, although the limited size of the resulting hydrolysis halos indicates low activity. Limited proteolytic action on skimmed milk was also detected in all *Trametes* sp. and *Lenzites* sp. isolates, while with gelatin as the substrate, only *Trametes* sp. S1.OA.A_F2 produced a clear hydrolysis halo. The different ability to utilize the gelatin or the casein contained in skimmed milk could be due to the substrate specificity of the enzymes produced by the isolates as reported in [62,63].

Cellulases and starch-degrading enzymes produced the largest hydrolysis halos, recalling the importance of these enzymes in fungal nutrition and colonization [64,65]. *Trametes* sp. S1.OA.A_F2, *Trametes* sp. S2.OTC.C_F5, *Trametes* sp. S2.OA.A_F5 and *Lenzites* sp. S3.OA.B_F6 showed the most marked amylolytic and cellulolytic halos (Table 1) and were selected to further characterize their enzymes by in vitro assays. Concentrated supernatants were then used in liquid assays in the presence of soluble starch or carboxymethylcellulose to assess the optimal pH value for their amylases or cellulases, respectively. *Lenzites* sp. S3.OA.B_F6 produced the highest amylolytic activity that reached the value of 4.676 nkat mL^−1^ (Appendix A). A pH of 5 was found to be optimal, while at pH 4 and 7, the enzymatic activity decreased to 50 and 20%, respectively. *Trametes* sp. S1.OA.A_F2, *Trametes* sp. S2.OTC.C_F5 and *Trametes* sp. S2.OA.A_F5 produced cellulases and amylases in low quantities, and their activities ranged from 0.201 (pH 4) to 0.494 nkat mL^−1^ (pH 4) for cellulases and from 0.155 (pH 7) to 0.513 (pH 7) nkat mL^−1^ for amylases (Appendix A).

Various types of extracellular enzymes were produced by the Antarctic fungal isolates. Endophytic fungi are ubiquitous in plant tissue and usually possess the specific hydrolytic systems required for depolymerize vegetal biomass, thus enabling the colonization process. Endophytic and phytopathogenic fungi seem to have similar invasion mechanisms, and many known pathogenic species are being isolated as endophytic in a range of plants [66]. The extracellular complex consists of the hydrolytic enzymes responsible for polysaccharide degradation and the oxidative ligninolytic system, which degrades lignin and opens phenyl rings [67,68]. For these reasons, assessing the bioactive potentials of the endophytes can pave the way for eco-friendly applications as a source of enzymes. Endophytic fungi remain an unexplored group of filamentous fungi, and their potential as enzyme producers has not been deeply explored yet. Enzymes from endophytic fungi isolated in extremophilic environments could be economically relevant and very interesting for the production of industrial biocompounds [69,70]. As an example, taking into consideration their aptitude to hydrolyze the complex structure of lignocellulose and the production of lipases, amylases, and proteases, extremophile endophytic fungi may become new sources of industrially relevant enzymes, suitable for the degradation of lignocellulosic biomass in the production of fuel ethanol and other value-added products. To evaluate the real potential of these endophytic fungi as sources of industrial biocatalysts, their physicochemical properties should be further characterized. Moreover, extensive studies of cultivation techniques, submerged or solid-state, are needed at both a laboratory and industrial scale to raise enzyme yields and make the production economically attractive.

### 3.3. Total Phenolic and Flavonoid Content of Fungal Extracts

Fungi represent a recognized source of natural products, crucial to health and with wide applications. A multitude of bioactive compounds, such as polysaccharides, terpenes, polyphenols, lectins, peptides, and proteins, have been isolated and exploited for their biological properties [71,72]. The high and increasing number of fungal genome sequences demonstrated that their biosynthetic potential is still underestimated since many pathways involved in the biosynthesis of secondary metabolites are silent under standard cultivation conditions [73]. Hence, due to the limitations in fungal cultivability and the lack of effective methods of exhaustive isolation of metabolites, the exploitation of fungi as a source of potential bioactive products is still scarce. Therefore, it is very important to characterize fungi living in peculiar life conditions such as in Antarctica, since environmental stressors as well as their host could induce the activation of normally cryptic pathways under different growth conditions [74]. Aqueous and methanol extracts were obtained from three Antarctic fungal isolates, i.e., *Trametes* sp. S2.OA.C_F6, *Lenzites* sp. S3.OA.B_F6 and *Sistotrema* sp. S1.OA.C_F2, and the total phenolic (TPC) and flavonoid (TFC) contents were measured due to their importance as signal molecules known to respond to changing environments. The two solvents used have been shown to give the highest yield in the extraction procedure. In particular, alcohols were proved the best solvents for phenol extraction [75]. The TPC was expressed in terms of gallic acid equivalent per gram of dry weight (GAE g^−1^) (standard curve equation: y = 1.1872x − 0.0235, R2 = 0.9956). The values obtained for the concentration of total phenols ranged from 1.0857 ± 0.011 mg GAE g^−1^ for the aqueous extract of *Sistotrema* sp. S1.OA.C_F2 to 6.0580 ± 0.062 mg GAE g^−1^ for the methanol extract of *Trametes* sp. S2.OA.C_F6 (Figure 2A and Appendix A).

Flavonoids, ubiquitous in plants and present in fungi, are a group of secondary metabolites derived from the phenylpropanoid pathway. In plants, they have important physiological roles in signaling and in response to changing environments. These compounds are widely studied for their diverse and important pharmacological activities such as antiviral, antimicrobial, antiallergic, antibacterial, anti-inflammatory, antifungal, cardioprotective, antiangiogenic, antithrombotic, antiproliferative, antitumor, antimalarial, apoptosis-inducing and antioxidant activities [76]. Flavonoids also play an important role in the prevention of degenerative diseases associated with oxidative stress [77,78]. Several data suggested the role of flavonoids in the establishment of plant roots endosymbiosis with certain fungi [79]. The total flavonoid content of the Antarctic fungal isolates was expressed as quercetin equivalent per gram of dry weight (QE g^−1^) (standard curve equation: y = 0.8688x − 0.0112, R2 = 0.9946), as reported in Figure 2B and Appendix A. The methanol extract of *Sistotrema* sp. exhibited the lowest concentration of flavonoids (1.309904 ± 0.014 mg QE g^−1^), while the highest was found in the aqueous extract of *Lenzites* sp. (5.51562 ± 0.022 mg QE g^−1^). Indeed, the aqueous extraction produced the highest yield of both TPC and TFC in *Lenzites* sp. compared to the other fungi. Conversely, methanol was found to be more effective in extracting both phenols and flavonoids from *Trametes* sp. compared to the other fungi. Furthermore, our results showed that *Sistotrema* sp. produced lower amounts of both TPC and TFC regardless of the solvent used.

### 3.4. Cancer Cells Appear to Be Sensitive to Fungal Extracts

The relatively high phenolic and flavonoid contents of the fungal extracts make these Antarctic species an interesting source of bioactive compounds for medicinal applications. The presence of phenols and flavonoids in plants or microorganisms was often correlated to antioxidant activity. In human organisms, the imbalance between reactive oxygen species (ROS) and the antioxidant system may cause oxidative stress, leading to cellular damage and subsequently to various diseases such as neurodegenerative disorders, aging, and cancer [80]. To verify whether our fungal extracts could impact human cell growth, we investigated their effect on the viability of two cell lines, one from human neuroblastoma (SH-SY5Y, tumor cells) and another from human lung epithelial cells (HBEC, non-tumor cells). As shown in Figure 3A,B, the extracts inhibited the growth of both cell lines in a dose-dependent manner, and the effect increased slightly over time. Moreover, the organic extracts showed greater cytotoxicity than the aqueous ones in all samples, which could be due to a higher phenol content found in all extracts [81]. Furthermore, all extracts appeared to exert a greater effect on cancer cells. *Trametes* sp. S2.OA.C_F6 methanol extract showed the strongest growth inhibitory activity for both tumor and non-tumor cell lines (IC50 = 1.69 ± 0.12 mg mL^−1^, IC50 = 2.42 ± 0.26 mg mL^−1^, respectively). On the other hand, as for the aqueous extracts, the most cytotoxic was that from *Lenzites* sp. S3.OA.B_F6 for both cell lines (IC50 = 1.89 ± 0.18 mg mL^−1^, IC50 = 15.76 ± 0.95 mg mL^−1^, respectively). Interestingly, other authors demonstrated that, by using the MTT assay, the ethanol extract of *Lenzites betulina* strongly reduced the viability of several types of cancer cells, and this effect appeared to be induced through ROS generation, reduction of the matrix metalloproteinases, P53 up-regulation, and Bcl2, pro-caspase-3, pro-caspase-9 and proline-rich acidic protein down-regulation [82]. The cytotoxic effect of our extracts appeared to be more intense than that of extracts derived from non-Antarctic species, although tested on different cell lines [82]. Interestingly, a positive correlation between TPC and the ability to inhibit cell growth was observed for all extracts regardless of the types of used cells.

### 3.5. Fungal Extracts Affect the Formation of Amyloid Fibrils

Parkinson’s disease is a neurodegenerative disorder associated with the amyloid aggregation of the presynaptic protein α-synuclein (Syn) and its familiar mutants [83]. Although several molecular events underlying Syn aggregation are still unclear, oxidative stress is known to contribute to this process. Some metabolites extracted from different fungi showed neuroprotective effects in vivo [84]. Polyphenols, pure or present in crude extracts, could protect against neurodegenerative disorders interfering with the oxidative stress mechanism in the brain and/or by reducing the aggregation propensity of Syn [85,86,87]. In this study, we analyzed the effects of the methanol extract of *Trametes* sp. S2.OA.C_F6 on the fibrillation process of Syn and its familiar mutant E46K. This extract was selected for its high content of polyphenols. Both proteins were incubated at 37 °C under shaking to promote protein fibrillation, in the presence or absence of the fungal extract. As assessed by Thioflavin T (ThT) assay (Figure 4A), amyloid fibrils were present in both Syn and its mutant already after 72 h of incubation, and their amount increased over time. At this stage, the protein secondary structure underwent conformational change (Figure 4B). Indeed, according to their unfolded nature [88], the proteins exhibited typical CD spectra of random coil conformation (black lines) at time 0, while adopting β-sheet secondary structure after incubation. As shown in Figure 4B, the β-sheet conformation is highlighted as a minimum at 218 nm in CD spectra recorded at 72 h and is more pronounced after 168 h of incubation. TEM pictures revealed the presence of structures compatible with mature unbranched fibrils with a diameter of 6–10 nm and length of 1 μm in the samples incubated for 168 h (Figure 4C), as previously observed [87]. The intensity of the ThT fluorescence emission at 485 nm of Syn and E46K samples was strongly reduced when the proteins were incubated with the methanol extract of *Trametes* sp. (1:10, Syn (or E46K)/extract, *w*/*w*), indicating the absence of fibrillar material in the samples (Figure 4A). This result was confirmed by far-UV CD analysis (Figure 4B). In fact, analyzing CD spectra after 72 and 168 h of incubation in the presence of the fungal extract, a predominantly random coil conformation was detected in both samples, indicating the absence of the transition to β-sheet secondary structure and, therefore, the absence of amyloid fibrils (Figure 4B). Moreover, TEM analysis indicated the presence of amorphous aggregates with a morphology completely different from amyloid fibrils. In conclusion, *Trametes* sp. methanol extract strongly hampers the amyloidogenic process of Syn and its mutant, as previously observed by using small molecules of natural origin [87,89].

### 3.6. Antarctic Fungi Affect the Root Architecture in Arabidopsis and Tomato

Several studies have reported the beneficial effects of plant-associated microbial communities on plant growth and fitness, even in extreme environments [20,21,22,23,24,25,26]. The knowledge of the influence of Antarctic endophytic fungi on plants of agricultural interest is still in its infancy, but it deserves to be deepened for the peculiarity of extremophilic microorganisms. To test the effect of the Antarctic Basidiomycetes *Trametes* sp. S2.OA.C_F6 and *Lenzites* sp. S3.OA.B_F6 on plant growth, co-cultivation experiments were carried out with *Arabidopsis thaliana* Col-0 and tomato (*Solanum lycopersicum* L.-cv. AKRAI F1) seedlings. To discriminate between the effects triggered by volatile or soluble molecules, both split and contact interaction experiments were carried out, respectively. As for Arabidopsis, at 7 days post-inoculum (dpi), the direct contact interaction caused a significant shortening of the primary root length (Figure 5A,C) and a simultaneous increase of the growth of secondary roots, leading to an overall enhancement in root density as compared to the control. Moreover, both fungi promoted the development of the aerial part, inducing a statistically significant increase in plant fresh weight (Figure 5C). On the other hand, the inhibitory effect on primary root length was lost in the split interaction (Figure 5B,D), while the growth promotion of secondary roots and aerial part was maintained, as evidenced by the statistically significant increase in plant fresh weight (Figure 5D).

This result suggested that diffusible soluble molecules secreted by the fungi could inhibit primary root length, while fungal volatile organic compounds (VOCs) could induce secondary roots formation and aerial part development. In tomato, the presence of the furrow in the plates did not substantially influence the effect of the fungus on the considered plant parameters and the two fungal isolates showed an opposite behavior on primary root length, which was significantly inhibited by *Trametes* sp. S2.OA.C_F6 and enhanced by *Lenzites* sp. S3.OA.B_F6. Moreover, both fungi had a positive effect on plant fresh weight (Figure 6, all panels).

These results suggested that volatile signals would be the key factors during either *Lenzites* sp. S3.OA.B_F6– or *Trametes* sp. S2.OA.C_F6–tomato interactions. Interestingly, similar effects were reported for *Trichoderma*–Arabidopsis interaction, where plant growth enhancement was mainly attributed to fungus-emitted VOCs, especially terpenes, whereas root morphology alteration was principally ascribed to fungus-induced regulation of auxin distribution in apical and lateral roots [90,91,92]. It has also been reported that acidification of the culture medium by *Trichoderma* is able to induce auxin redistribution within Arabidopsis columella root cap cells, causing root tip bending and growth inhibition [93]. To test whether Antarctic isolates were able to induce acidification of the medium, we grew both fungi on PDA plates supplemented with bromophenol blue as a pH indicator. Interestingly, we found that they were able to strongly acidify the medium (Appendix A), and it could be hypothesized that Arabidopsis plants adapt the root apparatus to the acidification via lateral root production. In general, our results suggest that both Antarctic fungi behave similarly to *Trichoderma* when interacting with plants, establishing beneficial relationships with them. This discovery opens up new perspectives on the possible exploitation of these fungi for sustainable agriculture. To this end, the experiment will be scaled-up growing plants first in pots and then in the field. 

### 3.7. Antarctic Fungi Antagonize the Growth of Crop Plant Pathogenic Fungi

Co-culturing of fungi has been reported as a successful tool to induce the activation of cryptic pathways, which trigger the production of new fungal secondary metabolites with potential biotechnological applications [94]. Specifically, the antagonistic reaction can stimulate the biosynthesis of bioactive natural molecules in the interaction zone to counteract the spreading of the antagonist [95]. In this work, the ability of the Antarctic fungal isolates of *Trametes* sp. S2.OA.C_F6, *Lenzites* sp. S3.OA.B_F6, *Sistotrema* sp. S1.OA.C_F2 and *Peniophora* sp. S3.OTC.C_F1 to antagonize or inhibit the growth of phytopathogenic fungi, i.e., *Fusarium graminearum* and *Botrytis cinerea*, was investigated by growing fungi in co-cultures. The observed antagonistic reactions included overgrowth, contact inhibition, and distance inhibition. Specifically, in both co-cultures with *F. graminearum* (Figure 7A) and *B. cinerea* (Figure 7B), either *Trametes* sp. S2.OA.C_F6 or *Lenzites* sp. S3.OA.B_F6 showed a strongly pigmented mycelium in the interaction zone, accompanied by an antagonistic effect on the phytopathogen growth. This phenotype might be explained by the production of volatile or diffusible antifungal metabolites, which include both signaling and reacting compounds, by one or both isolates as a result of the interaction [96]. On the other hand, *Sistotrema* sp. S1.OA.C_F2 seemed to be almost completely overwhelmed by the phytopathogens, whereas *Peniophora* sp. S3.OTC.C_F1 was the only isolate displaying distance inhibition against *B. cinerea.* For a better interpretation of the results, it is worth mentioning that the growth rates of *Trametes* sp. S2.OA.C_F6 and *Lenzites* sp. S3.OA.B_F6 are similar to that of *F. graminearum* and *B. cinerea* (7–10 days to cover the whole 90 mm Petri dish), whereas *Peniophora* sp. S3.OTC.C_F1 and *Sistotrema* sp. S1.OA.C_F2 grow slower than the phytopathogenic fungi (around 15 and 20 days to cover the whole 90 mm Petri dish, respectively), and thus, their antagonistic reaction could be slightly impaired.

Previous works reported that extracellular alkalinization contributes to fungal pathogenesis against both plants and microbes and acidification could reverse pathogenicity [97,98]. Thus, to understand the behavior of co-cultured fungi with respect to their ability to influence the pH of the growth medium, we also carried out the co-culturing experiments on PDA plates supplemented with bromophenol blue (Appendix A). The ability of *Trametes* sp. S2.OA.C_F6 and *Lenzites* sp. S3.OA.B_F6 to acidify the medium (Appendix A) could be hypothesized as one of the mechanisms adopted by the two fungal isolates to counteract the phytopathogen spreading. As for the other fungal isolates, none of them seemed able to acidify the medium (Appendix A). Examining the antagonistic potential of Antarctic fungi against important phytopathogens is the initial step in using these isolates for applications in plant biocontrol programs used in agriculture.

In conclusion, the search for novel bio-products in poorly explored environments is emerging as the key to providing solutions for many relevant issues, and Antarctic environments are increasingly recognized as precious locations for bioprospecting. This study emphasizes the ecological and biotechnological relevance of fungi from the Antarctic ecosystem. The fungal isolates used in this work have proved to be valuable sources of bioactive molecules ranging from industrial to pharmaceutical and agricultural applications, even though more studies are necessary to allow their biotechnological exploitation.

## Figures and Tables

**Figure 1 jof-08-00979-f001:**
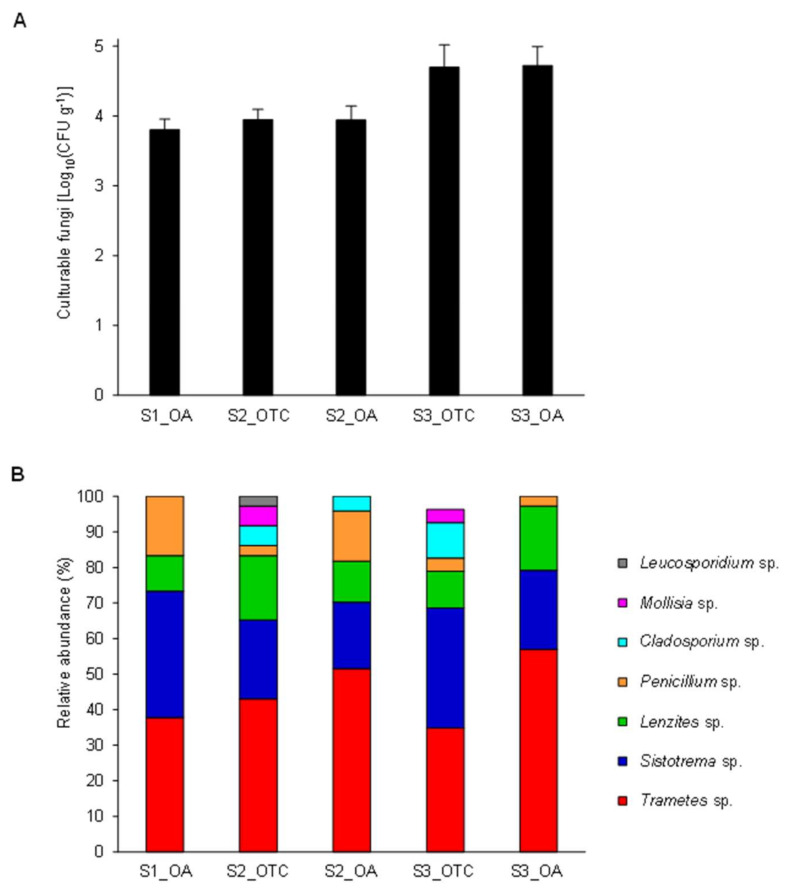
Endophytic fungi from *Colobanthus quitensis* leaves. (**A**) Colony-forming units (CFU) of culturable endophytic fungi per fresh plant weight unit (g) were evaluated for *C. quitensis* plantlets harvested in open areas (OA samples) as well as inside open-top chambers (OTC samples). Culturable fungi were grown at 25 ± 1 °C for 21 days on potato dextrose agar (PDA) supplemented with 0.25% lactic acid. No additional colonies could be observed for a longer incubation time. Mean Log_10_ (CFU g^−1^) and standard deviation values from three replicates (each as a pool of plants) are presented for each sample. (**B**) Relative abundances of taxa of culturable endophytic fungi assessed by morphological analyses. No significant differences were found among the collection sites (OA and OTC), according to Tukey’s test (*p* > 0.05).

**Figure 2 jof-08-00979-f002:**
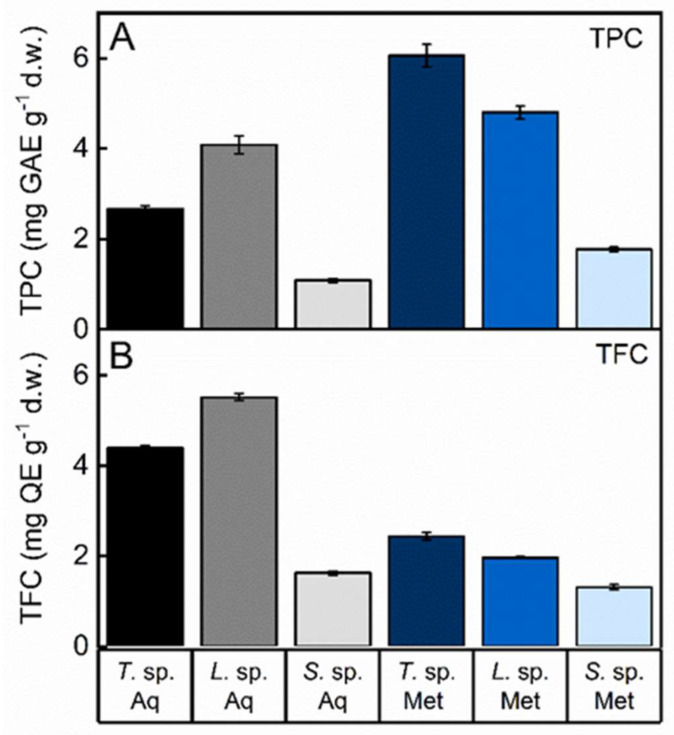
Total phenolic content (TPC) (**A**) and total flavonoid content (TFC) (**B**) of the aqueous (Aq) and methanol (Met) fungal extracts of *Trametes* sp. S2.OA.C_F6 (*T*. sp.), *Lenzites* sp. S3.OA.B_F6 (*L*. sp.) and *Sistotrema* sp. S1.OA.C_F2 (*S*. sp.). TPC was expressed in terms of gallic acid equivalent (GAE) g^−1^ dry weight; TFC was expressed as quercetin equivalent (QE) g^−1^ dry weight. Data were reported as mean values ± standard deviation (SD) of triplicates.

**Figure 3 jof-08-00979-f003:**
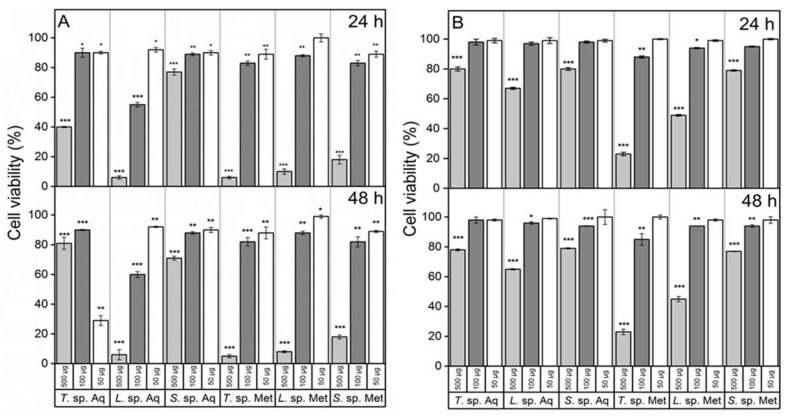
Cell viability test. SH-SY5Y (**A**) and HBEC (**B**) cells exposed for 24 and 48 h to increasing concentrations (50, 100, 500 µg) of aqueous (Aq) or methanol (Met) fungal extracts of *Trametes* sp. S2.OA.C_F6 (*T*. sp.), *Lenzites* sp. S3.OA.B_F6 (*L*. sp.) and *Sistotrema* sp. S1.OA.C_F2 (*S*. sp.). Results were expressed as a percentage of the corresponding untreated cultures. Error bars indicate the standard error of three independent experiments carried out in triplicate. Asterisks indicate statistically significant differences: *, *p* < 0.05; **, *p* < 0.01; ***, *p* < 0.001.

**Figure 4 jof-08-00979-f004:**
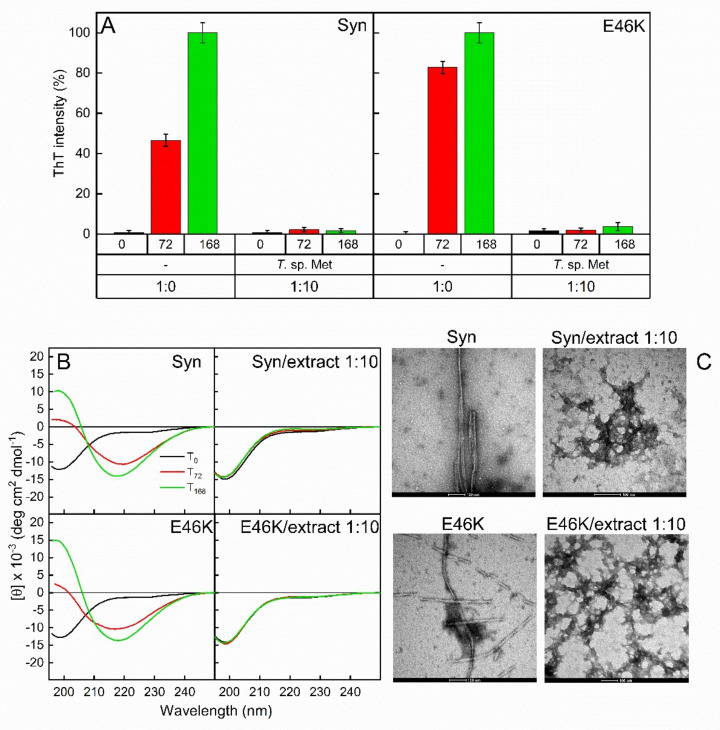
Amyloid formation inhibition probed by ThT fluorescence assay (**A**), circular dichroism (CD) (**B**) and TEM analysis (**C**). α-Synuclein (Syn) and its familial mutant (E46K) were left to aggregate in the absence (1:0) and in the presence (1:10) of methanol extract of *Trametes* sp. S2.OA.C_F6 (*T*. sp. Met). Aliquots taken from the mixtures were collected after 0 (black), 72 (red) and 168 h (green) of incubation. The TEM scale bar in every picture corresponds to 100 nm. Error bars indicate the standard error of three independent experiments carried out in triplicate.

**Figure 5 jof-08-00979-f005:**
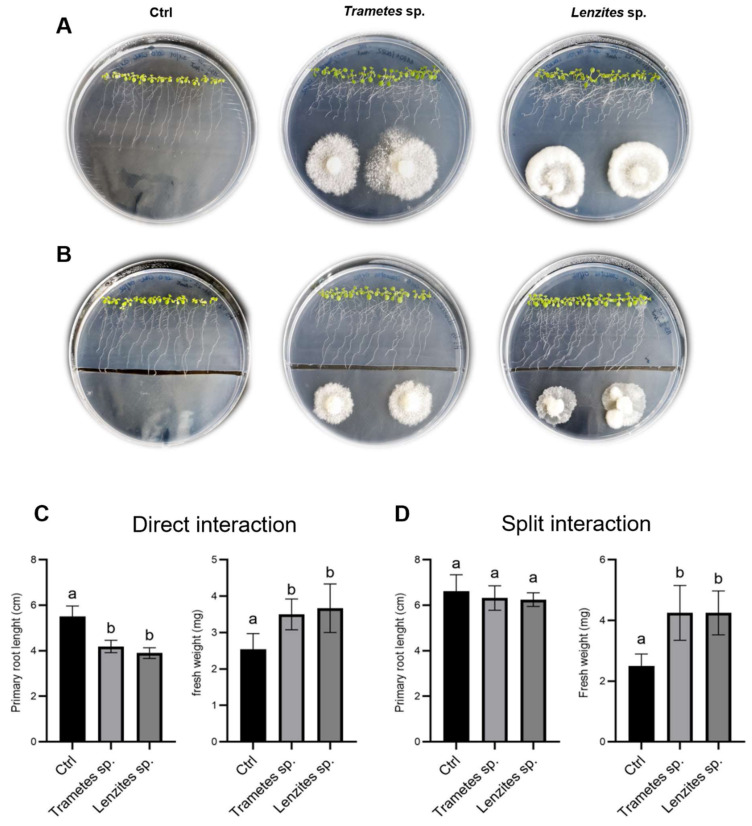
Effect of *Trametes* sp. S2.OA.C_F6 and *Lenzites* sp. S3.OA.B_F6 isolates on Arabidopsis growth. (**A**,**B**) Representative photos of 12-day-old Arabidopsis (Col-0) plantlets grown on MS½ medium at 7 dpi. (**A**) Direct interaction experiments and (**B**) split interaction experiments. (**C**,**D**) Physiological parameter measurements. Primary root length (cm) and fresh weight (mg) measured in direct interaction experiments and (**C**) split interaction experiments (**D**). Bar plots represent mean ± SD (*n* = 15). Different letters indicate significant differences (one-way ANOVA, *p* < 0.05; Tukey post hoc test, *p* < 0.05).

**Figure 6 jof-08-00979-f006:**
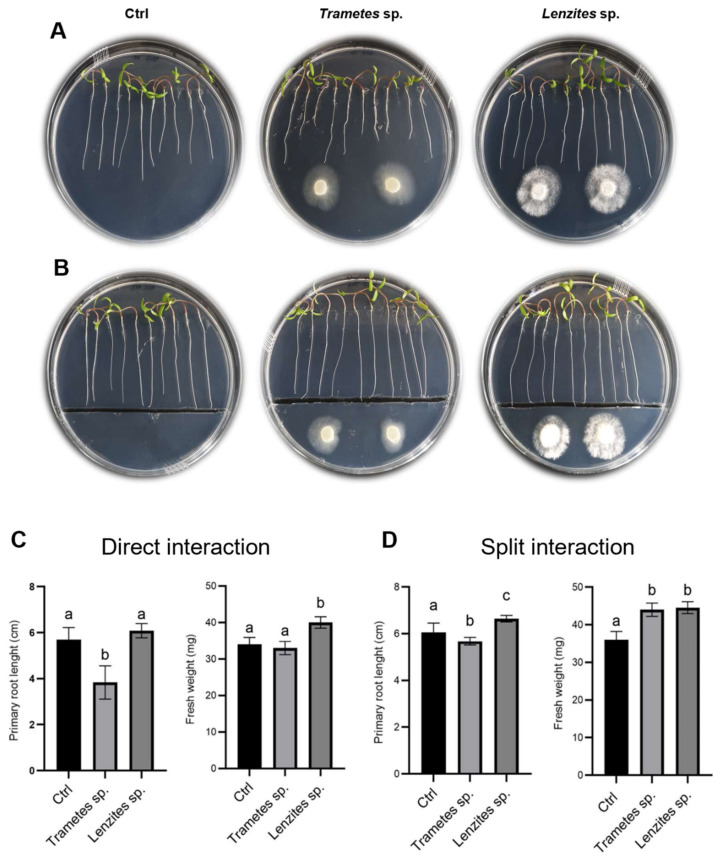
Effect of *Trametes* sp. S2.OA.C_F6 and *Lenzites* sp. S3.OA.B_F6 isolates on tomato growth. (**A**,**B**) Representative photos of 8-day-old tomato plantlets grown on MS½ medium at 3 dpi. (**A**) Direct interaction experiments and (**B**) split interaction experiments. (**C**,**D**) Physiological parameter measurements. Primary root length (cm) and fresh weight (mg) measured in direct interaction experiments (**C**) and split interaction experiments (**D**). Bar plots represent mean ± SD (*n* = 15). Different letters indicate significant differences (one-way ANOVA, *p* < 0.05; Tukey post hoc test, *p* < 0.05).

**Figure 7 jof-08-00979-f007:**
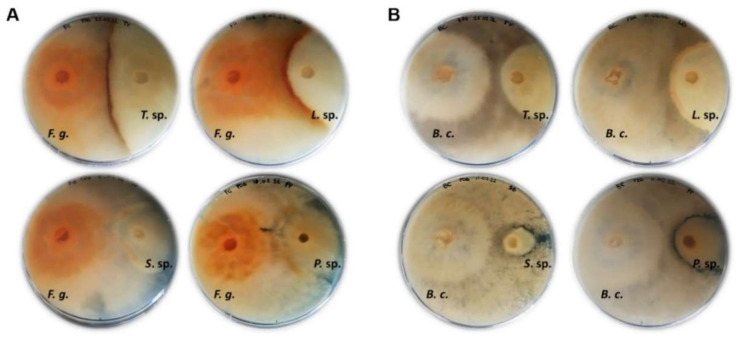
Co-cultivation of *Fusarium graminearum* (**A**) and *Botrytis cinerea* (**B**) (on the left of the plates) and Antarctic fungal isolates on PDA solid medium at 10 dpi. *F. g*.: *Fusarium graminearum*; *B. c*.: *Botrytis cinerea*; *T*. sp.: *Trametes* sp.; *L*. sp.: *Lenzites* sp.; *S*. sp.: *Sistotrema* sp.; *P*. sp.: *Peniophora* sp.

**Table 1 jof-08-00979-t001:** Screening for extracellular enzymatic activities of newly isolated Antarctic fungal isolates. Qualitative extracellular enzyme activity was determined by the appearance of haloes (diameter in mm) around the colony as follows: −, no halo; +, 5–20 mm halo; +++, ≥30 mm halo.

Fungal Isolate	Activity					
	Amylolytic	Lipolytic	Cellulolytic	Pectinolytic	Proteolytic (Skimmed Milk)	Proteolytic (Gelatin)
*Trametes* sp. S1.OA.A_F2	+	+	+++	+	+	+
*Trametes* sp. S2.OTC.C_F5	+	+	+++	+	+	−
*Trametes* sp. S2.OA.A_F5	+++	+	+	+	+	−
*Trametes* sp. S2.OA.C_F6	+	+	−	+	+	−
*Lenzites* sp. S2.OTC.C_F8	−	+	−	+	+	−
*Lenzites* sp. S3.OA.B_F6	+++	+	+	−	+	−
*Sistotrema* sp. S1.OA.C_F2	+	−	+	−	−	−
*Peniophora* sp. S3.OTC.C_F1	−	−	+	−	−	−

## Data Availability

Fungal ITS sequences presented in this study are available in NCBI (http://www.ncbi.nlm.nih.gov; accessed on 1 June 2022); accession numbers for each fungal isolate are indicated in Appendix A.

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
