# Peer review of "Biodiversity and Bioprospecting of Fungal Endophytes from the Antarctic Plant Colobanthus quitensis"

_jof, 2022, doi:10.3390/jof8090979_

Round 1
Reviewer 1 Report
It is an excellent and interesting result. However, I consider that the plant growth experiment using Petri plates is weak. An experiment using pots would be more suitable.
Reviewer 2 Report
The manuscript reports very interesting research on the biodiversity and bioprospecting of fungal endophytes from Colobanthus quitensis the plant growing in the Antarctica. We should look for novel bio-products in different environments. Extreme areas such as Antarctica are recognized having great potential. The paper prove it and provides an extensive data material and conclusions. Thus, results have great utility value in my opinion. They can be used primarily in industrial, pharmaceutical and agricultural applications. Undoubtedly, more research is needed for those endophytes exploitation.
It should be emphasized, that the paper is very well written, which shows very good organization, readability and grammar. I have noticed only few minor mistakes, e.g. Figure S2: an abbreviation “T. sp.” should be provided for Trametes sp.
In my opinion, the manuscript deserve to be published in Journal of Fungi, an MDPI journal.
Reviewer 3 Report
In the submitted manuscript by Bertini et al., entitled “Biodiversity and bioprospecting of fungal endophytes from the Antarctic plant Colobanthus quitensis” the authors isolated the fungal endophytic community associated with the leaves of the Antarctic plant Colobanthus quitensis and screened for bioactive compounds. Authors show that several fungal isolates exhibited extracellular enzyme activity, and their extracts displayed antitumor activity as well as inhibited the formation of amyloid fibrils on cell lines. Moreover, few fungal isolates showed positive growth effects in Arabidopsis and tomato.
Overall, the experiments included in the manuscript are informative to a certain extent. The manuscript writing is understandable and easy to read, though there were several instances where the transition is not appropriate between the consecutive sentences. In general, the manuscript presents evidence that fungal isolates from this work can be used as a source of bioactive molecules in future applications. However, as I detail below, there are several problems with the manuscript experiments and figures that must be addressed.
Fig. 1: Not very clear what the authors meant by “Culturable fungi were grown at 25±1°C for 21 days on Potato Dextrose Agar (PDA) supplemented with 0.25% lactic acid. No fungi were isolated by incubation on PDA plates at 15±1°C and no additional colonies were observed for a longer incubation time.” Whether the bar graph represents cfu at 25±1°C and not at 15±1°C? Please clearly mention it on the graph.
It would be more informative to include pictures of enzymatic activity assays as a supplementary figure.
Fig. 3: What do you mean by untreated culture? Not treated at all or treated with the solvents alone (methanol or aqueous)? Please include the control with solvent treatment only, to rule out the effect of methanol on cell viability.
Fig.3: What does the error bar represent? SD or SEM? How many replicates? statistical analysis? Please provide the information in the figure legend.
Fig. 4: Again solvent treated control is missing for amyloid formation inhibition assay. Also what does the error bar represents in fig 4A?
Reviewer 4 Report
The authors present the isolation and characterization of fungal endophytes from Colobanthus quitensis, an Antarctic plant. Interestingly they find Basidiomycetes, which have been scarcely been studied as endophytes. The work is well presented and most of the experiments and conclusions are sound and novel.
However some points should be addressed before it can be published:
Line 120: I presume that lactic acid was used to acidify the medium to avoid bacterial growth, Is this correct? Why the authors did not used antibiotics such as cloramphenicol or ampicillin to avoid bacterial colonies? Or did they observed bacterial growth and simply did not collect them? and if so, What is then the purpose to add lactic acid?
Line 144-145: BLASTn analysis of only one marker gives a very poor resolution to identify fungal species. If the authors want a more precise taxonomic annotation, they should at least perform a phylogenetic tree using type material ITS sequences, and better use more than one genetic marker (calmodulin, RNApol2, Beta tubulin, etc.). With this identification process authors should be careful on how they report the "genera" found in the NCBI, otherwise this can create confusion to other authors that use these sequences to perform their own identification analysis
Line 154: please italicize "in vitro"
Line 165: ten to the sixth? Is the super index missing in this number?
Line 180: as measured in mm of the halo diameter? This would give a semi-quantitative measure
Line 282: Did the growing speeds of the fungi were considered for these experiments? Reading the whole paper this is explained in the Results and Discussion section, but I think it would be friendlier to the reader to state the fact that the confronted fungi have different speed growth rates.
Line 309: It is possible that some isolates could be siblings (especially those isolated from the same site and source, although in a different replica (Table S1)? Reading further on this seems unlikely due to the differential pattern of enzymatic activity, Maybe it should be discussed here, Something like:
"The possibility of having isolated siblings from the replicas seems unlikely due their differential enzymatic pattern expression (see section 3.2 below)"
Line 315 (Figure legend): I suggest
..."could be..."
(instead of ..."were").
Or "No growth was observed..."
Otherwise it seems that it may have been growth but the authors decided not to isolate them.
This fact is strange since they were isolated at very low temperatures, Any idea of why at low temperatures no fungi were detected?
Line 399: It would have been interesting to test the sturdiness of the enzymes to heat, cold, ionic force, etc... compared with those obtained form Trametes sp. from a mesophile environment.
Just an idea for the next paper
Line 404: polysaccharides are not low molecular weight compounds, neither do proteins such as lectins.
Line 412: ...and as endophytes, this is also a very different condition since the plant may be modulating the production of secondary metabolites from the fungus.
See for example: Identification of a Huperzine A‑producing endophytic fungus from Phlegmariurus taxifolius. Molecular Biology Reports. 47, 489–495 (2020) https://doi.org/10.1007/
ITS2 ribotyping, in vitro anti-inflammatory screening, and metabolic profiling of fungal endophytes from the Mexican species Crescentia alata Kunth. South African Journal of Botany 134 (2020) 213-224 s11033-019-05155-1
Line 476: This is a very interesting result but it is not clear if the "untreated cells" mentioned in the Materials and Methods section simply were not added with anything, or if they were added with the same amount of solvent in which the extracts were obtained. That control is mandatory because the effect of methanol alone, for example, could be responsible for cell death or less growth
Line 488: italicize "in vivo"
Line 515: These experiments would be more conclusive if suitable controls had been performed: It is not clear in what solvent was the Trametes extract added, (methanol?) a control with solely the solvent is needed, since the solvent alone could prevent fibrillation. Also, a positive control, any substance that has been previously demonstrated that inhibits fibril formation would have also supported better these results.
Line 568: italicize Lenzites
Line 602: These experiments can be standardized by inoculating the slowest growing fungus a few days before inoculating the other fungus on the other side of the Petri dish. In this way, they should concur at the middle of the dish, or, if the inhibition is very strong, one of the fungi might stop growing and the effect can then be attributed solely to the inhibition and not the growth rate.
Round 2
Reviewer 1 Report
I still consider that pot experiments could be suitable. The authors must be included the limitation of the work.
Author Response
Q1. I still consider that pot experiments could be suitable. The authors must be included the limitation of the work.
R1. We thank the Reviewer and we strongly consider his suggestion, as we also think that pot experiments would help to thoroughly characterize the potentiality of the Antarctic fungi on improving commercial plants’ growth and fitness. To point out this statement, we added a phrase in the manuscript (lines 614-615). Nevertheless, in our experimental plan, this would be the second step, after verifying that Antarctic fungi are not detrimental for tomato and Arabidopsis development. The experiment carried out on Petri dishes showed that the selected Antarctic fungi are promising as plant growth promoters and thus we may plan pot experiments with more confidence. Moreover, the experiment on Petri dishes gave us important information on the influence of the Antarctic fungi on plant roots development and on the effect of the split and direct interactions on plant growth.
Reviewer 3 Report
I don't have any further comments.
Author Response
I don't have any further comments.
We thank the Reviewer for his approval.